# Green Manuring Enhances Soil Multifunctionality in Tobacco Field in Southwest China

**DOI:** 10.3390/microorganisms12050949

**Published:** 2024-05-07

**Authors:** Yu Feng, Hua Chen, Libo Fu, Mei Yin, Zhiyuan Wang, Yongmei Li, Weidong Cao

**Affiliations:** 1College of Plant Protection, Yunnan Agricultural University, Kunming 650500, China; 18787788824@163.com; 2Institute of Agricultural Environment and Resources, Yunnan Academy of Agricultural Sciences, Kunming 650205, China; chenhua19792003@126.com (H.C.); fulibo8888@163.com (L.F.); ymmay@163.com (M.Y.); wzyaas@163.com (Z.W.); 3College of Resources and Environment, Yunnan Agricultural University, Kunming 650500, China; 4State Key Laboratory of Efficient Utilization of Arid and Semi-Arid Arable Land in Northern China, Institute of Agricultural Resources and Regional Planning, Chinese Academy of Agricultural Sciences, Beijing 100081, China

**Keywords:** soil multifunctionality, green manure, smooth vetch, common vetch, networks, tobacco field

## Abstract

The use of green manure can substantially increase the microbial diversity and multifunctionality of soil. Green manuring practices are becoming popular for tobacco production in China. However, the influence of different green manures in tobacco fields has not yet been clarified. Here, smooth vetch (SV), hairy vetch (HV), broad bean (BB), common vetch (CV), rapeseed (RS), and radish (RD) were selected as green manures to investigate their impact on soil multifunctionality and evaluate their effects on enhancing soil quality for tobacco cultivation in southwest China. The biomass of tobacco was highest in the SV treatment. Soil pH declined, and soil organic matter (SOM), total nitrogen (TN), and dissolved organic carbon (DOC) content in CV and BB and activity of extracellular enzymes in SV and CV treatments were higher than those in other treatments. Fungal diversity declined in SV and CV but did not affect soil multifunctionality, indicating that bacterial communities contributed more to soil multifunctionality than fungal communities. The abundance of Firmicutes, Rhizobiales, and Micrococcales in SV and CV treatments increased and was negatively correlated with soil pH but positively correlated with soil multifunctionality, suggesting that the decrease in soil pH contributed to increases in the abundance of functional bacteria. In the bacteria–fungi co-occurrence network, the relative abundance of key ecological modules negatively correlated with soil multifunctionality and was low in SV, CV, BB, and RS treatments, and this was associated with reductions in soil pH and increases in the content of SOM and nitrate nitrogen (NO_3_^−^-N). Overall, we found that SV and CV are more beneficial for soil multifunctionality, and this was driven by the decrease in soil pH and the increase in SOM, TN, NO_3_^−^-N, and C- and N-cycling functional bacteria.

## 1. Introduction

Soil multifunctionality refers to the capacity of soil ecosystems to perform multiple ecological functions, including nutrient cycling, soil nutrient retention, soil nitrogen transformation, and litter decomposition [1,2]. Diverse microbial communities are considered to play key roles in the maintenance of soil multifunctionality. For example, microbial communities contribute to important ecological processes, such as decomposition and nutrient cycling [3]. Highly diverse microbial communities promote the soil nitrogen cycle. Various microorganisms are required for the degradation of organic matter [4], and bacterial diversity can affect the abundance of fungal pathogens [5].

Abiotic factors can also have an indirect effect on soil multifunctionality by altering biodiversity levels. For example, no tillage can enhance microbial diversity by increasing the content of soil organic carbon [6]. Li et al. [7] also showed that crop rotation can alter soil bacterial richness and community composition, which increases soil multifunctionality primarily by regulating carbon cycle-related enzyme activities. Other studies have shown that intercropping can enhance the content of available nutrients, which can increase the biodiversity of soil ecosystems [8]. Inorganic or organic fertilization can significantly affect the abundance and distribution of soil microbes [9]. Large amounts of fertilizers have been applied to meet the demand for crop growth, and this has resulted in a negative relationship between soil biodiversity, soil function, and crop productivity [10]. The excessive application of fertilizers over long periods can lead to alterations in soil biodiversity, community composition, and the loss of soil biodiversity. Several studies have shown that the application of organic fertilizers is an effective and sustainable agricultural practice [11,12,13].

Green manure is used as an effective measure for enhancing soil organic matter and mitigating soil degradation, and is a nice way to enhance soil multifunctionality [14]. Results from a long-term fertilizer experiment showed that green manure can promote soil nitrogen cycling by enhancing biological nitrogen fixation [15,16]. In addition, green manure promotes increases in the abundance of beneficial microbes and suppresses the growth of pathogenic microbes, resulting in the improvement of soil multifunctionality [5]. However, the contribution of green manure to soil functionality varies among green manure species. For example, leguminous green manures have a strong symbiotic nitrogen fixation ability and provide sufficient nitrogen for the soil, whereas green manure from cruciferous species can activate phosphorus or potassium in the soil and thereby increase the content of available nutrients [17,18]. Thus, it is necessary to clarify the effects of various green manures on soil multifunctionality.

In southwest China, tobacco (*Nicotiana tobacum* L.) cultivation has been one of the most important economic pillars of industry [19]. Long and continuous tobacco production has led to serious problems, such as soil nutrient imbalance, decreased organic matter content, and increased harmful metabolites in the soil. In addition, the long-term and extensive application of physiological acidic fertilizers causes soil acidification in tobacco plants, resulting in short main roots and fewer fibrous roots, making it difficult to absorb nutrients and reducing fertilizer utilization efficiency [20,21,22,23]. In the last two decades, smooth vetch has been introduced as winter green manure to improve and maintain soil quality in tobacco fields. In fact, there are some other winter green manures suitable for this area. However, the effects of various green manures on tobacco production have not yet been clarified. In this study, six green manure species were selected to explore their effects on soil multifunctionality at three growth stages of tobacco in the soil of a tobacco field. Our aim was to provide an insight into the role of different green manure options for improving soil quality in tobacco cultivation.

## 2. Materials and Methods

### 2.1. Experimental Design

A pot experiment was performed in Houxiang village, Longjie Town, Chengjiang City, Yunnan Province (24°39′29.654″ N, 102°53′29.418″ E, 1753 m above mean sea level). The pot experiment is a continuation of a previous field experiment, conducted under controlled conditions to facilitate monitoring and sampling. The field experiment is a rotation trial involving different types of green manure and tobacco that began in 2019. In October 2019, green manure crops were sown without any fertilizer application during their growth period, and they were tuned over at the end of March 2020. Tobacco seedlings were transplanted at the end of April. The fertilization quantities for tobacco were N 90 kg ha^−1^, P 45 kg ha^−1^, and K 270 kg ha^−1^. The tobacco was harvested in September 2020. Then, the soil would be prepared for the next rotation cycle to be carried out. Before the pot experiment commenced, the green manure–tobacco rotation field experiment was conducted for three years.

The pot experiment was carried out in the open air at the local tobacco test site. This was to ensure that the growth process of the potted tobacco was at the same climate level as the tobacco growing in the field. Topsoil from the 0–20 cm soil layer was used in the experiment, the control soil was sourced from winter fallow fields, and the other soil came from tobacco green manure rotation fields. The soil type belongs to the ferralsols. Its texture is sandy, and the soil particle composition consists of 59.7% sand, 26.4% silt, and 13.9% clay. The main properties of the original soil were as follows: pH, 7.34; soil organic matter (SOM), 35.1 g/kg; total nitrogen (TN), 2.52 g/kg; total phosphorus (TP), 2.60 g/kg; total potassium (TK), 16.3 g/kg; alkali-hydrolyzed nitrogen, 205 mg/kg; available phosphorus (AP), 97.5 mg/kg; available potassium (AK), 446 mg/kg; chloride ion (Cl), 52.2 mg/kg; bulk density, 1.14 g cm^−3^; average moisture content, 24%.

The experiment comprised seven treatments, and there were nine replicates in each treatment. Considering the possibility that tobacco plants might die during growth due to diseases or poor environmental adaptability, we included an additional replicate. The layout of the pot trial was randomly arranged. Winter fallow was used as the control (CK). The other six treatments were green manure species commonly grown in Yunnan: smooth vetch (SV) (*Vicia villosa* Roth var. *glabrescens*); hairy vetch (HV) (*Vicia villosa*); broad bean (BB) (*Vicia faba*); common vetch (CV) (*Vicia sativa* L.); rapeseed (RS) (*Brassica campestris*); and radish (RD) (*Raphanus sativus*). The pots were 40 cm tall, had an upper diameter of 35 cm, and held 20 kg of soil. The tobacco variety was K326.

Pot loading was completed on 27 April 2022. Before loading the soil into the pots, we broke down large soil blocks using a hammer, then sieved the soil to remove small stones, weeds, and other contaminants. Next, we thoroughly mixed the soil with a shovel and loaded the pots to ensure the uniformity of the soil in each pot as much as possible. Green manure crops had been harvested in mid-March from the previous field trial site. Considering there was over a month between harvesting the green manure and transplanting the tobacco seedlings, after we had harvested the green manure, we took a certain amount and killed it at 105 °C, then oven-dried it at 70 °C until a constant weight was reached. In order for it to be evenly distributed with the soil in the pots, we used tools to process the dried green manure into 3 cm-small pieces, which was more conducive to its decomposition. The green manure was mixed at a ratio of 1:500 [24,25] (i.e., 20 kg soil: 40 g green manure) and then placed into pots. Flue-cured tobacco seedlings were transplanted on 29 April 2022. Each pot had the same amount of fertilizers applied, i.e., N 360 g, P 180 g, and K 1080 g. Fertilizers were applied three times: 40% on the day of transplanting, 20% on the 15th day after transplanting, and 40% on the 30th day after transplanting. For the first fertilizer application, the fertilizer was mixed with the soil and put into the pot. For the second and third application, the fertilizer was dissolved in water and evenly poured into each pot. After fertilization, we conducted a test by slowly and evenly pouring 5 L of water into a pot containing 20 kg of soil. A small amount of water drained from the vent hole at the bottom of the pot. We established this as the standard watering procedure and subsequently poured 5 L of water into each experimental pot. On the 45th, 65th, and 90th days after transplantation (i.e., 45DAT, 65DAT, and 90DAT of the growth period of tobacco), three tobacco plants were randomly sampled for each treatment and soil samples were collected at the same time to facilitate the validation of the repeatability of the results and statistical processing of experimental errors. Since our sampling method was destructive, it involved carefully removing entire plants and soil from each pot during each sampling event. Rhizosphere soil samples were collected following a previously described method [26]. After the tobacco plants had been completely extracted, the soil attached to the roots was gently shaken, and the part of the soil tightly bound to the root surface (i.e., the rhizosphere soil) was gently brushed down using a sterile brush (one brush per treatment). The rhizosphere soil was then mixed thoroughly, passed through a 2 mm sieve, and stored at −80 °C for DNA extraction. Non-rhizosphere soil samples were also taken and evenly mixed, and were divided into two parts: 80 g of fresh soil samples used for analysis of soil enzyme activities, and the remaining soil was naturally air-dried and then passed through the 0.149 mm sieve for the physical and chemical analysis. Tobacco roots and shoots were separated, washed, then killed at 105 °C for 30 min and dried at 70 °C for more than 8 h to constant weight to determine their biomass.

### 2.2. Analysis of Soil Physical and Chemical Properties

The content of soil organic matter (SOM) was measured using the oil bath heating potassium dichromate volumetric method. The content of total nitrogen (TN) was measured using the Kjeldahl method. The content of total phosphorus (TP) was measured using HF-HClO_4_ digestion–molybdenum blue colorimetry, and the content of total potassium (TK) was measured using flame photometry (Shanghai AOPU, AP1302, Shanghai, China). Dissolved organic carbon (DOC) was extracted from fresh soil using ultrapure water. The soil:water ratio was 1:2, and samples were shaken at 4 °C and 200 r/min for 2 h and centrifuged at 4 °C and 12,000 r/min for 15 min. The supernatant was passed through a 0.45 μm membrane filter, and measurements were taken using a TOC/N instrument (Multi N/C 2100, Analytik Jena, Jena, Germany). Mineral nitrogen (NH_4_^+^-N and NO_3_^−^-N) was extracted with 2 mol/L KCl (soil:water ratio 1:10) and tested using a continuous flow analyzer. Available phosphorus (AP) in the soil was extracted using ammonium fluoride hydrochloride, and the content was measured using the molybdenum blue colorimetric method. Rapidly available potassium (AK) was extracted using 1 mol/L ammonium acetate and determined using flame photometry (Shanghai AOPU, AP1302, China). The concentration of chloride ions (Cl) was determined using AgNO_3_ titration. The pH was measured using an electrode (water:soil ratio 1:1) (Mettler Toledo, FiveEasy Plus, Columbus, OH, USA). The procedures used to analyze extracellular enzyme activities related to the carbon, nitrogen, and phosphorus cycles in the soil are described in detail in Zhou Guopeng [27].

### 2.3. Soil DNA Extraction and PCR Amplification

The total DNA of the microbial community was extracted from the rhizosphere soil samples using an E.Z.N.A.^®^ soil DNA kit (Omega Biotek, Norcross, GA, USA) as per the manufacturer’s protocol. The quality of the extracted DNA was determined using 1% agarose gel electrophoresis, and the concentration and purity of DNA were evaluated using a Nanodrop 2000 spectrophotometer (Thermo Fisher Scientific, Waltham, MA, USA). PCR was performed to amplify the V3–V4 variable region of the 16S rRNA gene using the 338F (5′-ACTCCTACGGGGGGCAG-3′) and 806R (5′-GACTACHVGGGTWTCTAAT-3′) primers, and the ITS region was amplified using the ITS1F (5′-CTTGGTCATTTAGAGTAAA-3′) and ITS2R (5′-GCTGTTCGATGC-3′) primers. The thermal cycling conditions were as follows: 95 °C for 3 min (initial denaturation); 27 cycles of 95 °C for 30 s (denaturation), 56 °C (16S rRNA) and 50 °C (ITS) for 30 s (annealing), 72 °C for 30 s (extension); 72 °C for 10 min (final extension); and a holding temperature of 4 °C (ABI GeneAmp 9700 thermocycler, Marshall Scientific, Hampton, NH, USA). The 20 μL PCR reactions contained 5× TransStart FastPfu buffer (4 μL), 2.5 mM dNTPs (2 μL), forward and reverse primers (5 μM; 0.8 μL each), TransStart FastPfu DNA Polymerase (0.4 μL), 10 ng of DNA template, and the rest with nuclease-free water. Assays of all samples were performed in triplicate.

### 2.4. Illumina MiSeq Sequencing

PCR products from the same sample were mixed and recovered using 2% agarose gel electrophoresis. The recovered products were purified using an Axy-Prep DNA Gel Extraction Kit (Axygen Biosciences, Union City, CA, USA), and the purified products were detected using 2% agarose gel electrophoresis. The recovered products were quantified using a Quantus™ Fluorometer (Promega, Madison, WI, USA). Library construction was conducted using a NEXTflex™ Rapid DNA-Seq Kit (Bio Scientific, Austin, TX, USA): (1) adapters were ligated; (2) magnetic beads were used to remove self-ligated adapter fragments; (3) PCR amplification was used to enrich the library templates; and (4) magnetic beads were used to recover the PCR products to obtain the final library. Sequencing was conducted using the Illumina platform (Shanghai Majorbio Bio-pharm Technology Co., Ltd., Shanghai, China). Raw data were uploaded to the NCBI SRA database (accession number: PRJNA1082904).

### 2.5. Bioinformatic Analysis

Fastp software [28] (https://github.com/OpenGene/fastp (accessed on 10 January 2023) version 0.20.0) was used for quality control of the raw sequencing reads. FLASH software [29] (http://www.cbcb.umd.edu/software/flash (accessed on 10 January 2023), version 1.2.7) was used to merge reads: (1) bases with quality scores below 20 at the read tails were trimmed. A 50 bp sliding window was used, and bases from the start of the window were removed if the average quality score within the window was below 20, and reads shorter than 50 bp and reads containing N bases were removed; (2) according to the overlap between paired-end reads, reads were merged into a single sequence with a minimum overlap length of 10 bp; (3) the maximum mismatch ratio permitted in the overlapping region was 0.2, and sequences that did not meet this criterion were discarded; and (4) samples were demultiplexed according to the barcodes and primers at the start and end of the sequences, and the sequence directions were adjusted. No mismatches were permitted in the barcodes, and a maximum of two mismatches was permitted in the primers. UPARSE software (http://drive5.com/uparse/ (accessed on 11 January 2023), version 7.1) was used to group the sequences into amplicon sequence variants (ASVs) at 97% similarity [30,31], and chimeric sequences were removed. The RDP classifier [32] (http://rdp.cme.msu.edu/ (accessed on 11 January 2023), version 2.2) was used to obtain taxonomic annotations for each sequence by aligning them against the SILVA 16S rRNA database (version 138) and UNITE database, with a confidence threshold of 70%. In all soil samples, 84,585 bacterial ASVs and 10,022 fungal ASVs were acquired. To ensure that subsequent analyses were conducted at the same sequencing depth, the data were rarefied, and this resulted in 72,611 bacterial ASVs and 8894 fungal ASVs.

### 2.6. Data Analysis

#### 2.6.1. Soil Multifunctionality

Soil multifunctionality was quantified following procedures described by Fanin et al. [33]. First, Shapiro–Wilk tests were used to assess the normality of the distribution across various soil physicochemical parameters. Datasets that did not follow a normal distribution were log-transformed or square root-transformed to meet the assumptions of normality prior to standardization [34]. For datasets containing negative values, the minimum value was subtracted from all data points to convert them to a positive scale before standardization [35,36]. Standardized (Z-score) values for all indices related to soil nutrients (total and bioavailable N, P, K; NH_4_^+^-N and NO_3_^−^-N; DOC) and enzymatic activity, including the activity of α-glucosidase (AG), β-glucosidase (BG), β-cellobiosidase (BC), β-xylosidase (BX), acetylglucosaminidase (NAG), leucine aminopeptidase (LA), and phosphatase (PP), comprised the soil multifunctionality dataset.

#### 2.6.2. Microbial Co-Occurrence Network Analysis

The R package WGCNA (v.1.70-3) was used to conduct a weighted gene co-expression network analysis (WGCNA) on the soil microbial communities [37]. First, ASVs present in at least 80% of the soil samples were selected, and the filtered bacterial and fungal ASV abundance tables were merged into a new table containing 332 bacterial and 66 fungal ASVs. Pairwise correlations between ASVs were calculated according to the Spearman correlation matrix, and *p*-values were adjusted using the FDR method. ASV pairs with correlation coefficients less than 0.45 and *p*-values greater than 0.05 were removed. A bacterial–fungal co-occurrence network was constructed using all samples, and modularity analysis was conducted on the network using the Gephi platform. The overall network was divided into eight ecological modules. The relative abundance of the species within each ecological cluster was standardized (Z-score) and averaged to obtain the relative abundance of each ecological module [38]. We focused on the five most abundant ecological modules.

#### 2.6.3. Statistical Analysis

Statistical comparisons among treatments were conducted using one-way analysis of variance (ANOVA) conducted with the *aov()* function in R software (version 4.2.2; R Core Team, R Foundation for Statistical Computing, Vienna, Austria). Significant differences between means were identified by Fisher’s least significant difference (LSD) test at a significance level of *p* < 0.05. The alpha diversity of bacteria and fungi was analyzed by the vegan (2.6-4) package of R software, including the ACE index reflecting community richness and Shannon–Wiener index reflecting community diversity [39]. The principal coordinate analysis (PCoA) based on Bray–Curtis dissimilarities was calculated in R using the vegan package (version 2.6-4) [40]. Analysis of similarity (ANOSIM) and permutational multivariate analysis of variance (PERMANOVA) were used to evaluate the effects of different treatments on the composition of the microbial community. Linear discriminant analysis effect size (LEfSe) was used, and the linear discriminant analysis (LDA) score threshold was 3.5.

## 3. Results

### 3.1. Effects of Different Green Manures on Tobacco Biomass, Soil Properties, and Extracellular Enzyme Activity

Increases in flue-cured tobacco biomass varied among treatments at different growth stages. At 45DAT, differences among treatments were negligible. At 65DAT, differences between green manure treatments were apparent and biomass was highest in SV and BB treatments. At 90DAT, the biomass was 24.9–87.6% higher in the green manure treatments than in CK, and the highest biomass was found in SV treatment (Figure 1).

At 45DAT, compared with CK, green manure treatments reduced soil pH, with HV decreasing the most by 6%. The BB and RS treatments exhibited the most substantial increase in SOM content, with an average enhancement of over 20%. Similarly, the BB treatment demonstrated the highest increase in TN content, with a 35% increase. The TK and AK contents of RD treatment were the highest, increasing by 9.7% and 139%, respectively, compared to CK.

At 65DAT, HV had the lowest pH, which was 3.9% lower than CK, and SOM content changes were similar to 45DAT, with BB and RS having the highest increases. BB treatment resulted in the highest levels of TN, TP, and AP, with increases of 37%, 36.9%, and 112%, respectively. The changes in TK, DOC, NH4^+^-N, and NO_3_^−^-N treatments were not significant.

At 90DAT, SV, HV, CV, and RD treatments significantly decreased pH by 5–6.3%. BB and CV treatments showed the greatest increase in SOM and TN. Green manure treatments significantly increased TP (16.5–47.1%) and AP (58.3–196%). RD treatment had the highest TK and AK contents, increasing by 18.9% and 645.5%, respectively. CV had a significant increase in NH_4_^+^-N and SV in NO_3_^−^-N compared to CK.

In summary, different types of green manure exhibited varying effects on the physicochemical properties of tobacco field soil. Specifically, HV and SV had the greatest impact on soil pH. BB treatment was most effective in increasing SOM, TN, and TP contents. RD treatment had the most pronounced increases in TK and AK. SV treatment increased NO_3_^−^-N content the most (Table 1).

At 45DAT, the enzyme activity of each treatment did not change significantly, while at 65DAT, only LA and AP exhibited a slight increase. At 90DAT, the green manure treatments demonstrated a significant increase in enzyme activity compared to CK, with the SV and CV treatments displaying the highest enzyme activities among all treatments. This suggests that the green manures significantly enhanced nutrient cycling during tobacco maturity, with the SV and CV treatments demonstrating the most pronounced effects (Table 2).

### 3.2. The Composition and Structure of Rhizosphere Microbial Communities in Response to Different Green Manures

Different treatments and sampling dates did not have significant effects on the diversity (Shannon index) or richness (ACE index) of bacterial communities in rhizosphere soil (*p* > 0.05). Sampling date and green manure treatment had extremely significant effects on soil fungal richness (ACE index) (*p* < 0.01) Appendix A. This suggests that the fungal community in the rhizosphere soil of tobacco fields exhibits a more sensitive response to green manure incorporation compared to the bacterial community.

The Shannon index of fungal diversity was significantly lower in the SV and CV treatments than in the control (*p* < 0.05) (Figure 2a). Differences in fungal richness between sampling dates and treatments were pronounced. Fungal richness was significantly lower in the SV treatment than in CK at all sampling dates (*p* < 0.05). At 90DAT, fungal richness was also significantly lower in the CV treatment than in CK, suggesting that the changes in the soil environment induced by the SV and CV treatments were not conducive to fungal enrichment or growth (Figure 2b).

PCoA plots according to Bray–Curtis distance and the results of β-diversity analysis revealed significant differences in ASVs among treatments across all sampling dates for both bacterial and fungal communities in tobacco rhizosphere soil (Figure 3). This indicates that the composition of rhizosphere microbial communities in tobacco soil underwent substantial changes in response to the different green manure treatments.

A total of 46 bacterial phyla and 1263 bacterial genera were detected across the 63 soil samples. Twenty-five phyla were detected in over 80% of the samples examined. The main phyla, which collectively comprised more than 95% of all bacteria, were Proteobacteria (30.0%), Actinobacteria (27.7%), Acidobacteria (13.2%), Gemmatimonadetes (10.1%), Chloroflexi (7.4%), Bacteroidetes (5.1%), Verrucomicrobia (1.7%), and Rokubacteria (1.2%). The five most abundant genera detected were subgroup_6, g_unclassified_f_Gemmatimonadaceae, g_unclassified_f_Intrasporangiaceae, Gemmatimonas, and Sphingomonas (Appendix A).

A total of 15 fungal phyla and 471 fungal genera were detected. Eight to nine phyla were present in 80% of the samples. The five most abundant phyla comprised 98.9–99.9% of fungi across all samples from the three sampling dates, and these were considered the dominant phyla. These phyla included Ascomycota, Mortierellomycota, Olpidiomycota, and Basidiomycota. The main genera were Botryotrichum, Acremonium, Mortierella, Olpidium, Fusarium, Chaetomium, and Plectosphaerella, which accounted for more than 70% of all fungi. The composition of fungal species did not vary significantly among sampling dates. At 65DAT, the abundance of Penicillium was high and the abundance of Olpidium was low; at 90DAT, the abundance of Lecanicillium was high and the abundance of Plectosphaerella was low. Changes in the abundance of these taxa accounted for a relatively small proportion of the overall changes (Figure 4).

Analysis of the differential abundance of bacterial and fungal communities using linear discriminant analysis effect size (LEfSe) analysis revealed taxa that were significantly enriched in specific treatments. The cladogram identified bacterial taxa, such as Firmicutes, Bacilli, Bacillales, Bacillaceae and *Bacillus*, which were particularly abundant in the SV treatment, and the abundance of Rhizobiales and Micrococcales was higher in the CV treatment than in the other treatments. The abundance of Dothideomycetes, Pleosporales, Mucoromycota, and Microascales was significantly higher in the BB, RS, SV, and CV treatments, respectively, than in the other treatments (Figure 5). This result indicates that the soil environment treated with SV and CV is particularly conducive for the enrichment in beneficial bacteria such as *Bacillus* and Rhizobiales, but it is also beneficial for the growth of some fungi with similar environmental preferences.

### 3.3. The Effect of Different Green Manures on Soil Multifunctionality

Soil multifunctionality was significantly higher in the green manure treatments than in CK (Figure 6). Differences in soil multifunctionality in the green manure treatments increased with flue-cured tobacco growth. At the 90DAT stage, soil multifunctionality was highest in the SV and CV treatments, which is consistent with the results for extracellular enzyme activity. This indicates that enzyme activity may directly influence the enhancement of soil multifunctionality (Appendix A).

### 3.4. Soil Bacterial–Fungal Symbiotic Network Analysis under Different Green Manure Treatments

WGCNA was used to construct a bacterial–fungal symbiotic network based on all samples. Analysis of the network topology revealed eight ecological modules, and we focused on the five most abundant ecological modules (modules 0–4) (Figure 7a).

The relative abundance of ecological modules 0–4 varied among treatments (Figure 7b). Module 0 was the only module that did not exhibit any significant variation in abundance among treatments. The relative abundance of module 1 was lower and the relative abundance of module 4 was significantly higher in the SV treatment than in CK. The relative abundance of module 2 was significantly lower in the BB, CV, and RS treatments than in CK, and the relative abundance of module 3 was significantly lower in the SV, HV, BB, and CV treatments than in CK. Module 2 and module 3 were considered the key ecological modules due to their differences in relative abundance across treatments.

Analysis of the composition of the microbial community in the key ecological modules revealed that fungi were the most abundant microbes in module 2 and module 3 (Figure 8) (Appendix A). The most abundant microbes in module 2 were *Fusarium*, *Plectosphaerella*, and *Penicillium* from the phylum Ascomycota, and their relative abundance among fungi in this module was approximately 59%. The main bacteria in module 2 were Actinobacteria (25%), Proteobacteria (24%), Gemmatimonadetes (22%), Acidobacteria (17%), and Chloroflexi (4%) (Appendix A). The most abundant fungal genera in module 3 were *Talaromyces* and *Chaetomium*, and their relative abundance in this module was approximately 34%. The main bacteria in module 3 were similar to those in module 2 and included Actinobacteria (43%), Proteobacteria (14%), Gemmatimonadetes (13%), Acidobacteria (9%), Chloroflexi (7%), Bacteroidetes (6%), and Nitrospirae (1%) (Appendix A).

According to the functional prediction results from FAPROTAX (Figure 9a) and FUNGuild (Figure 9b), methylotrophic, fermentation, anaerobic chemoheterotrophic, and nitrate reduction functions were stronger in module 2 than in module 3. Aerobic nitrification was stronger in module 3 than in module 2. In addition, phytopathogenic and pathogenic symbiotic fungi were more abundant in module 2 than in module 3. The microorganisms within and across ecological modules were mainly engaged in synergetic interactions (Appendix A).

Mantel tests revealed that soil pH and NO_3_^−^-N are key environmental factors affecting the relative abundance of modules 2 and 3 (*p* < 0.05) (Figure 10a). Linear regression analysis of the relative abundance of these two modules with environmental factors was performed. Module 2 was negatively and significantly correlated with soil TN and SOM content, and module 3 was positively correlated with soil pH and negatively correlated with soil NO_3_^−^-N content (*p* < 0.05) (Figure 10b).

Regression analysis revealed that module 2 and module 3 were significantly and negatively correlated with soil multifunctionality, while other ecological modules were not significantly correlated with soil multifunctionality (Figure 11).

## 4. Discussion

### 4.1. Effect of Green Manure Application on Soil Nutrients and Extracellular Enzyme Activities

The application of green manure led to increases in the soil’s SOM, TN, DOC, TK, and AK content and decreases in soil pH (Appendix A). Green manure is rich in organic matter, which decomposes in the soil via the activities of microbes. This releases DOC and nutrients such as nitrogen, but also P. This decomposition process increases the content of SOM, which is key for maintaining the structure and fertility of soil. As the organic matter decomposes, the soil releases protons due to nitrification, which causes a decrease in pH. In addition, the organic acids produced during the decomposition of green manure can lead to a decrease in soil pH [41]. The incorporation of potassium from green manure into the soil enhances TK and AK levels. In this experiment, the pH was more than 6% lower in the SV and CV treatments than in CK, especially in the late stage of tobacco growth (90DAT). The pH in both of these treatments was close to 6.5, which is within the range of values known to be most suitable for tobacco quality (5.5–6.5) [42]. This slight acidification can promote an increase in the availability of nutrients by mediating the dissolution of phosphates and trace minerals. The high extracellular enzyme activity in each treatment indicates that the application of green manure increases nutrient conversion efficiency, and the activity of extracellular enzymes was highest in the SV and CV treatments.

### 4.2. Response of Microbial Diversity to Different Green Manures

The α diversity of bacteria was not significantly affected by green manure treatment; however, green manure treatment had a significant effect on β diversity. Given that variation in soil microbial β diversity is derived from differences in microbial community composition, microbial community β diversity is often significantly associated with soil functions [43]. Firmicutes, *Bacilli*, *Bacillales*, and *Bacillus* were highly abundant in the SV treatment, and Maicrococcales and Rhizobiales were highly abundant in the CV treatment. *Bacillus* belongs to Firmicutes, which comprises typical plant growth-promoting rhizobacteria. Most of the species in this genus can perform various functions, including phosphorus solubilization, nitrogen fixation, protein degradation, and lignin degradation. Rhizobiales can establish symbiotic relationships with leguminous plants. It can also fix nitrogen in the atmosphere and convert it into inorganic nitrogen forms, such as nitrate and ammonium, which can be absorbed and utilized by plants. *Agrogenes* in the order Micrococcales performs nitrogen fixation functions. These microbes thus play important roles in the nitrogen cycle [44,45,46,47]. Regression analysis indicated that these differentially abundant species were significantly positively correlated with soil multifunctionality (Appendix A). Soil multifunctionality was highest in the SV and CV treatments, which might stem from the significant enrichment in these functional bacteria in these treatments. Regression analysis was conducted to clarify the relationships between the main environmental factors and these differentially abundant species. the abundance of these species were negatively correlated with soil pH and positively correlated with the content of soil TN, SOM, NO_3_^−^-N, and DOC. This suggests that SV and CV treatments decreased soil pH and increased the content of TN, SOM, NO_3_^−^-N, and DOC, which promoted increases in the abundance of functional bacteria.

Patterns of fungal diversity differed from patterns of bacterial diversity. Fungal diversity and richness were significantly lower in the SV and CV treatments than in the other treatments. Fungal diversity and richness did not significantly vary among the other treatments. However, the reduced fungal diversity and richness in the SV and CV treatments did not have any effect on the improvement in soil multifunctionality in these treatments. This might stem from the high redundancy of the metabolic functions of the microbial community [48], i.e., each metabolic function can be performed by multiple coexisting and taxonomically distinct microorganisms, which mitigates the deleterious effects of biodiversity decreases on soil function [49]. Thus, the reduction in fungal community diversity has no significant effect on soil multifunctionality [50]. The abundance of Mucoromycota and Microscales was significantly higher in the SV and CV treatments than in the other treatments, but their abundance were not significantly correlated with soil multifunctionality (Appendix A). This suggests that bacterial communities contribute more to soil multifunctionality than fungal communities. The observed decrease in diversity and richness might stem from the fact that fungi tend to be more sensitive to environmental changes than bacteria during the decomposition of plant organisms [51,52].

### 4.3. The Effect of Different Green Manures on Network Modules

Soil microbial co-occurrence network analysis is an important approach for evaluating the relationships between soil microorganisms and relationships between network characteristics and soil functions [53,54]. The abundance of modules 2 and 3 was significantly negatively correlated with soil multifunctionality. The abundance of module 2 was significantly lower in the BB, CV, and RS treatments than in the other treatments, and the abundance of module 3 was lower in the SV, HV, CV, and BB treatments than in the other treatments. This suggests that leguminous green manures had a stronger inhibitory effect on the microbial community in the modules than the other manures. The relative abundance of fungi was highest in modules 2 and 3 (Appendix A), but their trophic modes differed.

*Fusarium* was the dominant genus in module 2, and members of this genus are pathogenic, saprotrophic, or symbiotic. This was followed by *Plectosphaerella*, which comprises pathogenic fungi. Members of the genus Fusarium are pathogens that cause diseases such as tobacco root rot, wheat head blight, and tomato wilt [55]. Members of the genus *Plectosphaerella* are pathogens causing *Plectosphaerella* blight in tomatoes [56]. The nodes in the bacterial–fungi symbiotic network were negatively correlated with ASV_39013 (*Lysobacteria*) in module 0 and ASV_55972 (MND1) in module 3 (Appendix A). *Lysobacteria* can secrete various extracellular enzymes, such as chitinase and β-1,3-glucanase, to inhibit the growth of pathogenic fungi by hydrolyzing their cell walls [57]. MND1 belongs to the family Nitrosomonadaceae and is an important group involved in nitrification [47], suggesting that pathogens suppress soil functions by inhibiting functional microbial communities.

The most abundant fungal genus in module 3 was *Talaromyces*, which is an adaptable saprotrophic mold that produces highly active lignocellulolytic enzymes [58]. This genus engaged in antagonistic interactions with *Arenimonas* (ASV_57233) and *Adhaeribacter* (ASV_88935) (Appendix A). *Arenimonas* contributes to the N cycle by performing denitrification [59]. *Adhaeribacter* belongs to the phylum Bacteroidetes, and previous studies have found that the abundance of *Adhaeribacter* is negatively correlated with soil respiration [60]. Our results showed that functional microorganisms inhibited increases in soil multifunctionality, which stemmed from their competitive and antagonistic interactions.

Variation in the relative abundance of module 2 and module 3 in the different treatments might be attributable to soil physical and chemical properties. Soil pH is a major determinant of the effect of planting systems on microbial community structure [61,62]. Soil pH was significantly positively correlated and NO_3_^−^-N significantly negatively correlated with the relative abundance of module 3. However, soil pH was negatively correlated and NO_3_^−^-N positively correlated with soil multifunctionality (Appendix A). The addition of green manure, especially leguminous green manure, increases nitrification in the soil. In the leguminous green manure treatments (SV and CV), the pH was lowest in the later stage of tobacco growth, and NO_3_^−^-N was highest in these treatments. This indicates that soil pH and the NO_3_^−^-N content were the main environmental factors inhibiting the relative abundance of module 3. In addition, NO_3_^−^-N is the primary form of nitrogen absorbed by tobacco roots. NO_3_^−^-N is more conducive to the growth of tobacco plants than NH_4_^+^-N [42]. Favorable growth conditions promote increases in the abundance of beneficial microbial communities, which promotes the colonization of beneficial microbes and increases their abundance [63]. These beneficial microbes can engage in competitive and antagonistic interactions with pathogens, thereby reducing their relative abundance [64] and enhancing soil functionality. The abundance of module 2 was significantly negatively correlated with the SOM content (Appendix A). The SOM content was highest in the BB and RS treatments (Appendix A). Therefore, soil SOM is the main environmental factor limiting the relative abundance of module 2.

## 5. Conclusions

The results of our study indicate that green manure enhanced the biomass of tobacco, soil multifunctionality and microbial diversity, and the SV and CV treatments were the most effective. These improvements were primarily attributed to decreases in soil pH and increases in the content of SOM, TN, and DOC, which positively affect microbial communities and their functions. Our findings suggest that fungal diversity is not directly related to soil functionality; however, bacterial communities play a key role in enhancing soil multifunctionality. Bacterial communities with increased abundance in the SV and CV treatments, such as Firmicutes, Rhizobiales, and Micrococcales, can promote the cycling of C, N, and P nutrients, thereby enhancing soil multifunctionality. In the SV, CV, BB, and RS treatments, we observed a significant decrease in the relative abundance of network modules negatively correlated with soil multifunctionality. This reduction mainly stemmed from competition and inhibition between functional bacteria and pathogenic fungi or between bacteria and functional fungi. Environmental factors, especially variation in soil pH and the content of SOM and NO_3_^−^-N, inhibit increases in the abundance of these modules.

## Figures and Tables

**Figure 1 microorganisms-12-00949-f001:**
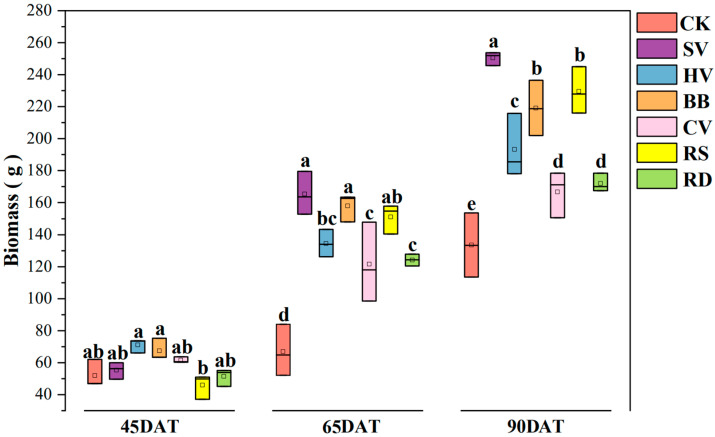
Tobacco biomass of different green manure treatments at 45, 65 and 90 days after tobacco transplanting. CK—winter fallow control; SV—returning smooth vetch; HV—returning hairy vetch; BB—returning broad bean; CV—returning common vetch; RS—returning rapeseed; RD—returning radish. Different lowercase letters on the top of boxes indicate significant differences among treatments at the same stage (*p* < 0.05); 45, 65, and 90DAT: 45, 65, and 90 days after transplanting, respectively. Data were collected during 13 June, 3 July, and 28 July in 2022 at Houxiang village, China.

**Figure 2 microorganisms-12-00949-f002:**
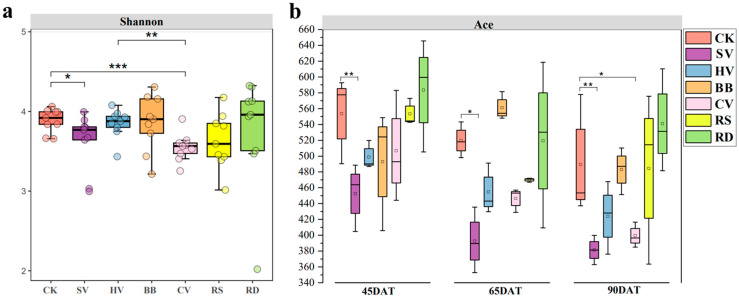
Analysis of the fungal community diversity of rhizosphere soil under different green manure treatments at different growth stages of tobacco. (**a**) Shannon index; (**b**) ACE index. CK—winter fallow control; SV—returning smooth vetch; HV—returning hairy vetch; BB—returning broad bean; CV—returning common vetch; RS—returning rapeseed; RD—returning radish. *, ** and *** indicate the effect at 0.05, 0.01 and 0.001 significant levels. 45, 65, and 90DAT: 45, 65, and 90 days after transplanting, respectively. Data were collected during 13 June, 3 July, and 28 July in 2022 at Houxiang village, China.

**Figure 3 microorganisms-12-00949-f003:**
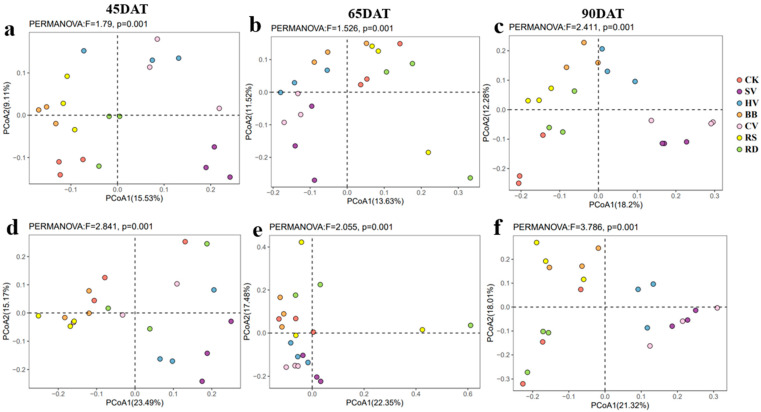
PCoA plots of the bacterial (**a**–**c**) and fungal (**d**–**f**) community in rhizosphere soil in the different treatments at growth stages of tobacco. CK—winter fallow control; SV—returning smooth vetch; HV—returning hairy vetch; BB—returning broad bean; CV—returning common vetch; RS—returning rapeseed; RD—returning radish. 45, 65, and 90DAT: 45, 65, and 90 days after transplanting, respectively. Data were collected during 13 June, 3 July, and 28 July in 2022 at Houxiang village, China.

**Figure 4 microorganisms-12-00949-f004:**
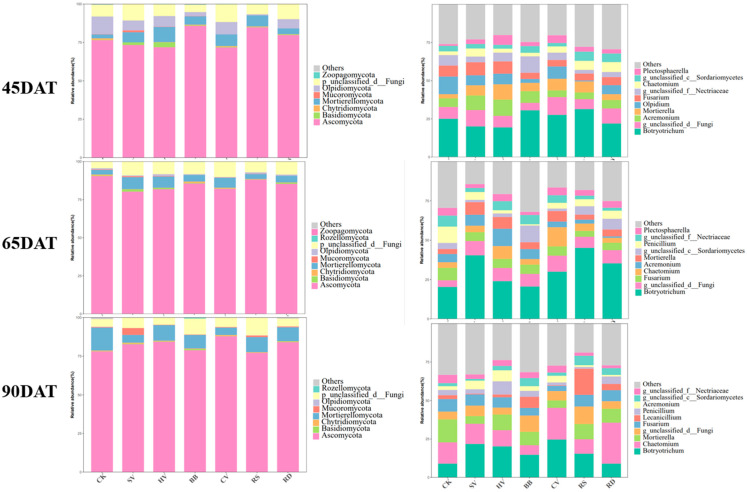
Composition of the rhizosphere soil fungal communities in different treatments across the three sampling dates. The left set of graphs shows phylum-level changes, and the right set of graphs shows genus-level changes. CK—winter fallow control; SV—returning smooth vetch; HV—returning hairy vetch; BB—returning broad bean; CV—returning common vetch; RS—returning rapeseed; RD—returning radish.

**Figure 5 microorganisms-12-00949-f005:**
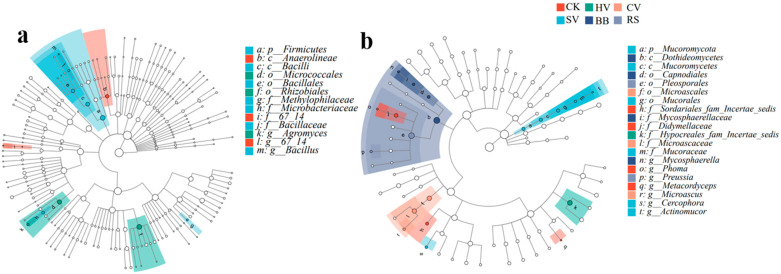
Taxonomic cladogram showing the results of LEfSe analysis of bacterial (**a**) and fungal (**b**) communities.

**Figure 6 microorganisms-12-00949-f006:**
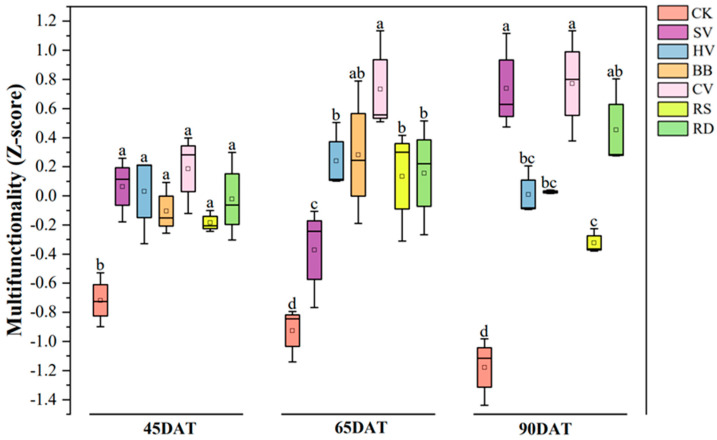
Development of soil multifunctionality as affected by six green manure options. CK—winter fallow control; SV—returning smooth vetch; HV—returning hairy vetch; BB—returning broad bean; CV—returning common vetch; RS—returning rapeseed; RD—returning radish. Different lowercase letters at the tops of boxes indicate significant differences among treatments at the same stage (*p* < 0.05); 45, 65, and 90DAT: 45, 65, and 90 days after transplanting, respectively. Data were collected during 13 June, 3 July, and 28 July in 2022 at Houxiang village, China.

**Figure 7 microorganisms-12-00949-f007:**
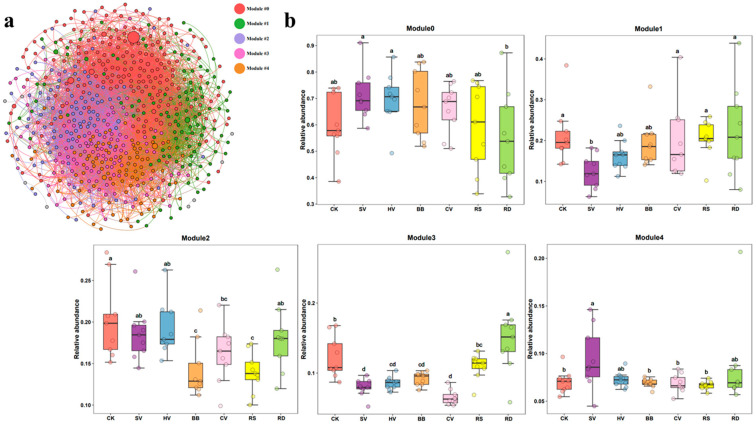
(**a**) Weighted gene co-expression network analysis (WGCNA) of bacterial–fungal interactions based on all samples; (**b**) relative abundance of ecological modules in the different treatments. CK—winter fallow control; SV—returning smooth vetch; HV—returning hairy vetch; BB—returning broad bean; CV—returning common vetch; RS—returning rapeseed; RD—returning radish. Different lowercase letters at the tops of boxes indicate significant differences among treatments at the same stage (*p* < 0.05).

**Figure 8 microorganisms-12-00949-f008:**
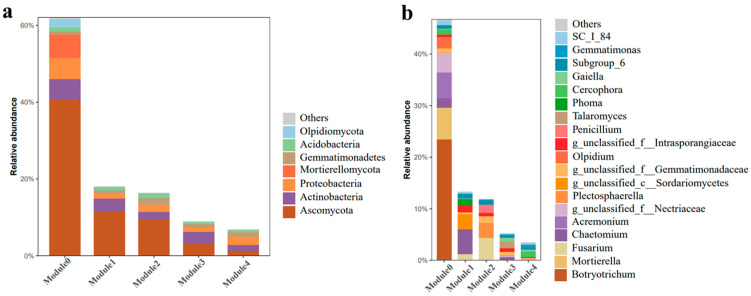
Microbial community composition of the co-expression network modules at the (**a**) phylum and (**b**) genus levels.

**Figure 9 microorganisms-12-00949-f009:**
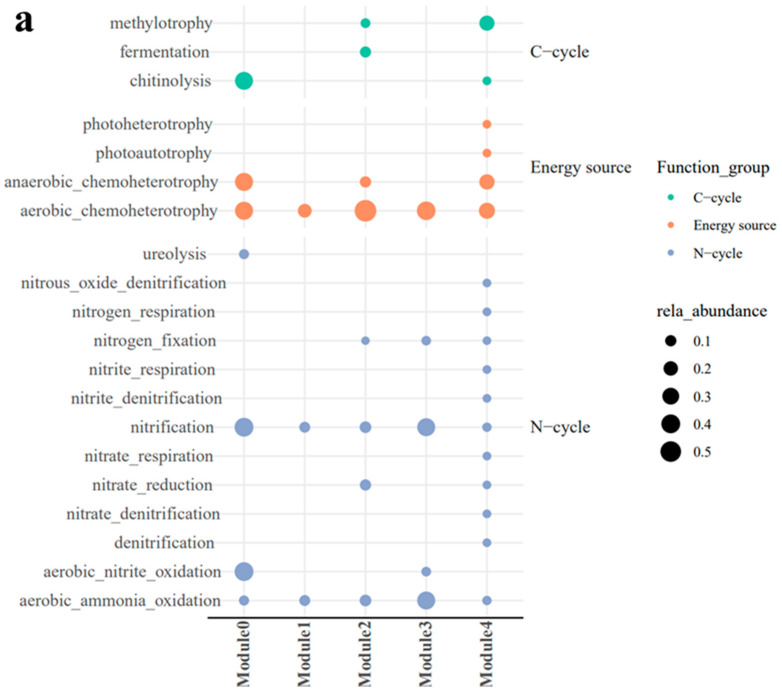
FAPROTAX (**a**) and FUNGuild (**b**) analysis of ecological modules, with black dots indicating the relative abundance of bacterial and fungal ASVs in each functional group (metabolic pathway).

**Figure 10 microorganisms-12-00949-f010:**
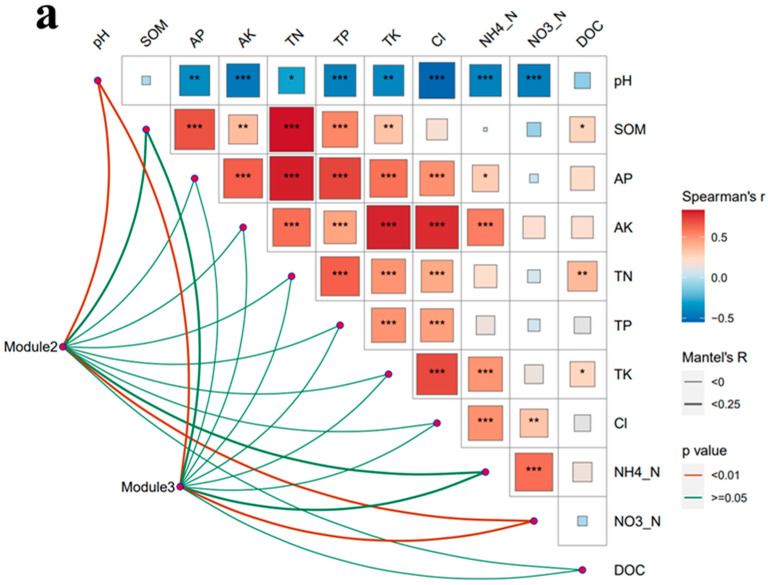
(**a**) Environmental factors affecting the abundance of key ecological modules identified through Mantel tests. (**b**) Linear regression analysis of the relative abundance of modules 2 and 3 with environmental factors. *, ** and *** indicate the effect at 0.05, 0.01 and 0.001 significant levels.

**Figure 11 microorganisms-12-00949-f011:**
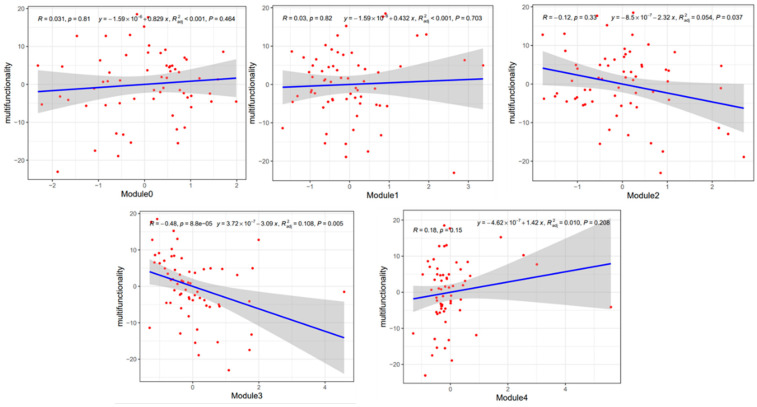
Regression analysis of soil multifunctionality and ecological modules.

**Table 1 microorganisms-12-00949-t001:** Soil physicochemical properties of treatments at three growth stages of tobacco.

Stage	Treatment	pH	SOM (g kg^−1^)	TN (g kg^−1^)	TP (g kg^−1^)	TK (g kg^−1^)	DOC (mg kg^−1^)	NH_4_^+^-N (mg kg^−1^)	NO_3_^−^-N (mg kg^−1^)	AP (mg kg^−1^)	AK (mg kg^−1^)
45DAT	CK	6.82 a	36.47 d	2.20 e	1.73 c	9.19 b	44 ab	1.62 ab	28.5 ab	88 e	750 b
	SV	6.6 bc	38.53 c	2.50 d	2.02 b	9.73 ab	32.1 ab	1.72 ab	51.5 ab	118.5 cd	1516.7 a
	HV	6.41 d	41.34 b	2.79 b	2.31 a	9.7 ab	41.1 ab	2.21 ab	38.2 ab	156 ab	1436.7 a
	BB	6.49 cd	45.59 a	2.97 a	2.26 a	10 ab	41 b	1.11 b	27 b	174 a	1550 a
	CV	6.69 ab	41.5 b	2.72 bc	1.97 b	9.29 ab	36.5 a	4.8 a	39.7 a	143.17 bc	1530 a
	RS	6.69 ab	44.27 a	2.74 b	2.02 b	9.63 ab	42.7 b	0.26 b	20 b	123.83 cd	1303.3 a
	RD	6.65 bc	41.82 b	2.59 cd	2.17 ab	10.08 a	35.5 ab	1.84 ab	42.2 ab	107.83 de	1790 a
65DAT	CK	6.90 ab	36.1 d	2.24 e	1.79 c	8.82 a	34.6 a	0.05 b	11.1 b	79.2 e	576.7 e
	SV	6.63 cd	38.3 c	2.48 d	1.91 c	9.18 a	49.5 a	0.33 ab	38.3 ab	84.8 de	903.3 de
	HV	6.52 d	40.9 b	2.64 c	2.24 ab	9.37 a	36.7 a	2.59 a	51.3 a	125.3 bcd	1013.3 cde
	BB	6.74 bc	45.4 a	3.07 a	2.45 a	9.93 a	46.1 a	0.3 ab	23.5 ab	168.2 a	1556.7 ab
	CV	6.67 cd	43.2 a	3.02 a	2.23 ab	9.85 a	50.1 a	1.08 ab	42.2 ab	154.8 ab	2050 a
	RS	6.77 bc	45 a	2.83 b	2.06 bc	9.99 a	37 a	0.97 ab	29.3 ab	127.3 abc	1520 bc
	RD	6.94 a	40.7 b	2.66 c	2.03 bc	8.86 a	43.9 a	1.35 ab	46.3 ab	88.2 cde	1420 bcd
90DAT	CK	6.94 a	34 e	2.06 c	1.55 d	7.77 d	29.3 b	0.03 c	11.1 c	47.2 d	188.7 c
	SV	6.5 b	38.2 d	2.52 b	1.95 bc	8.85 ab	33.7 b	0.77 ab	124.7 a	76.5 c	910 b
	HV	6.58 b	41 bc	2.47 b	2.28 a	8.42 bc	36.9 b	0.35 bc	32.2 bc	118.8 b	836.7 b
	BB	6.94 a	45.8 a	2.8 a	2.27 a	8.22 cd	35 b	0.18 c	28 bc	139.8 a	413.3 c
	CV	6.53 b	42 b	2.84 a	2.07 abc	8.67 bc	50.6 a	0.87 a	46.7 bc	118.2 b	1066.7 b
	RS	7.12 a	44.1 a	2.54 b	1.8 cd	7.89 d	35.3 b	0.13 c	32.6 bc	74.7 c	269.3 c
	RD	6.56 b	39.7 cd	2.59 b	2.16 ab	9.24 a	30.1 b	0.41 abc	58.8 b	115.5 b	1406.7 a

Note: Values are means. Values in the same column followed by different lowercase letters differed significantly at *p* < 0.05 (LSD test). SOM, soil organic matter; TN, total nitrogen; TP, total phosphorus; TK, total potassium; DOC, dissolved organic carbon; NH_4_^+^-N, ammonium nitrogen; NO_3_^−^-N, nitrate nitrogen; AP, available phosphorus; AK, available potassium. CK—winter fallow control; SV—returning smooth vetch; HV—returning hairy vetch; BB—returning broad bean; CV—returning common vetch; RS—returning rapeseed; RD—returning radish. 45, 65, and 90DAT: 45, 65, and 90 days after transplanting, respectively. Data were collected during 13 June, 3 July, and 28 July in 2022 at Houxiang village, China.

**Table 2 microorganisms-12-00949-t002:** Extracellular enzyme activity (nmol h^−1^ g^−1^) in soil of treatments at three growth stages of tobacco.

Stage	Treatment	AG	BG	BC	BX	NAG	LA	AP
45DAT	CK	3.72 a	26.9 ab	3.27 ab	3.00 b	6.65 b	−84.01 c	22.3 ab
	SV	6.14 a	45.9 a	5.53 ab	6.29 a	17.4 a	3.83 b	59.9 a
	HV	3.61 a	26.3 ab	3.11 b	3.41 ab	11.1 ab	12.78 b	57 ab
	BB	4.11 a	19.8 b	2.14 b	2.53 b	9.10 b	−78.88 c	19.9 b
	CV	6.25 a	51.6 a	7.88 a	5.99 a	16.7 a	97.73 a	43 ab
	RS	4.7 a	29.3 ab	3.72 ab	4.45 ab	13.4 ab	5.35 b	28.7 ab
	RD	3.65 a	26.9 ab	3.03 b	3.44 ab	11 ab	9.93 b	35.6 ab
65DAT	CK	4.11 a	27.2 b	3.9 b	3.73 ab	9.2 b	35 b	28.9 c
	SV	2.31 a	32.8 b	3.33 b	3.11 b	18.9 ab	34.9 b	34.2 bc
	HV	4.87 a	49.9 ab	6.23 ab	6.18 ab	17 ab	202.7 a	106.8 a
	BB	6.8 a	38 b	6.3 ab	7.22 ab	14.5 ab	−9.2 b	73.6 abc
	CV	6.36 a	69.6 a	9.85 a	9.23 a	26.5 a	27.6 b	95.3 ab
	RS	5.92 a	44.3 ab	5.86 ab	6.09 ab	15.3 ab	265.8 a	34.7 bc
	RD	5.54 a	50.5 ab	7.97 ab	6.03 ab	24.6 a	200.3 a	43.2 abc
90DAT	CK	7.2 c	33.2 c	3.4 c	4.91 c	8.43 c	166.3 d	41.4 c
	SV	30.9 a	154 a	23.9 a	17.39 ab	40.8 ab	328.4 a	187.6 a
	HV	10.7 bc	51.6 c	7.1 bc	8.14 c	19.1 c	235 bc	81.6 c
	BB	12 bc	54.3 c	6.8 bc	8.24 c	19.4 c	206.4 d	109.1 bc
	CV	22.9 ab	109.4 ab	15.5 ab	21.49 a	42.2 a	239 bc	167 ab
	RS	13.3 bc	60.9 bc	7.7 bc	8.15 c	19.3 c	224.5 bcd	54.2 c
	RD	16.9 abc	72.2 bc	10.5 bc	11.16 bc	24 bc	240.4 b	105.6 bc

Note: Values are means. Values in the same column followed by different lowercase letters differed significantly at *p* < 0.05 (LSD test). AG, α-glucosidase; BG, β-glucosidase; BC, β-cellobiosidase; BX, β-xylosidase; NAG, N-acetyl-glucosaminidase; LA, leucine aminopeptidase; AP, acid phosphatase. CK—Winter fallow control; SV—returning smooth vetch; HV—returning hairy vetch; BB—returning broad bean; CV—returning common vetch; RS—returning rapeseed; RD—returning radish. 45, 65, and 90DAT: 45, 65, and 90 days after transplanting, respectively. Data were collected during 13 June, 3 July, and 28 July 2022 at Houxiang village, China.

## Data Availability

Raw sequencing data are available in the NCBI Sequence Read Archive (study ID PRJNA937072).

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
