# Peer review of "Green Manuring Enhances Soil Multifunctionality in Tobacco Field in Southwest China"

_microorganisms, 2024, doi:10.3390/microorganisms12050949_

Round 1

Reviewer 1 Report

Comments and Suggestions for Authors

Dear authors,

the manuscripts covers an interesting topic. However, important information is ommited and needs to be added to the manuscript before a final decision on the quality of the study can be made. Particularly the Materials and Methods section lacks details on how the pot experiment was carried.

Please find my comments in the file provided. If these points can be clarified, it is worth considering the manuscript for publication but not in its current version.

Comments on the Quality of English Language

The Englsih is fine; I made some suggestions how to improve in the file I provided along with the review.

Reviewer 2 Report

Comments and Suggestions for Authors

Final report microorganisms -2960435

Green manuring enhances the soil multifunctionality in tobacco field in south-west China

For the purpose of this investigation, six distinct green manures were chosen to investigate the impact that they have on the multifunctionality of the soil in tobacco planting areas. By gaining an understanding of the characteristics of the soil, the authors intended to develop green manures that are more suited for the cultivation of tobacco.

The current study found that green manure increases tobacco biomass, soil multifunctionality, and microbial diversity, with the most effective treatments being SV and CV. These gains were primarily due to lower soil pH and higher levels of SOM, TN, and DOC, which benefit microbial populations and their functions. The results indicated that fungal diversity is not directly connected to soil functioning; instead, bacterial communities play an important role in improving soil multifunctionality. Bacterial communities with higher abundances in the SV and CV treatments, such as Firmicutes, Rhizobiales, and Micrococcales, can boost C, N, and P nutrient cycling, increasing soil multifunctionality. In the SV, CV, BB, and RS treatments, the relative abundance of network modules decreased significantly, which was adversely linked with soil multifunctionality. This decline was mostly caused by competition and inhibition between functional bacteria and pathogenic fungi or between bacteria and functional fungi. Environmental influences, particularly variations in soil pH and the amount of SOM and NO3--N, limit the abundance of these modules.

The document you gave is not well-written and should be read by someone who speaks English.

Here are some comments I have to make in order to improve the paper:

Be careful not to use acronyms in the figure captions or the footnotes; instead, make sure each figure can be understood on its own. All figures must be self-explanatory.

Please ensure that all tables are self-explanatory and that any abbreviations are explained in full.

Despite the fact that all of the references are adequate, I recommend including further references from 2023 and 2024.

When I looked over the journals, I saw that some of them were written in their entirety, while others were written in a more condensed form. I have made a note of this in the references. The journal style requires that you stick to the format, which consists of either the full name of the journal or a condensed version of it. Make sure you look at the directions that have been given to the authors.

Noting the page numbers of the textbook is requested when providing references for textbooks. Furthermore, the city in which the publication is based should also be specified.

If all of the comments that were stated above were made by the authors, then this manuscript may be approved with significant changes. It will be necessary for me to make one more revision in order to guarantee that all of my suggestions have been incorporated.

Comments on the Quality of English Language

Extensive editing of English language required

Round 2

Reviewer 1 Report

Comments and Suggestions for Authors

Dear authors,

thank you for the revision. Nevertheless I have some minor points that need to be addressed. See my notes in the file uploaded.

Kind regards,

Comments on the Quality of English Language

Reviewer 2 Report

Comments and Suggestions for Authors

Accept 

Author Response

Dear Editor and anonymous reviewers,

Thank you for reviewing my manuscript. I appreciate your time and attention to it. I noticed that no specific comments were provided on my submission. We are more than willing to make revisions if there are any areas that require further clarification or improvement.